# Can Sports Practice in Childhood and Adolescence Be Associated with Higher Intensities of Physical Activity in Adult Life? A Retrospective Study in Community-Dwelling Adults

**DOI:** 10.3390/ijerph192214753

**Published:** 2022-11-10

**Authors:** Gabriela C. Silva, William R. Tebar, Italo R. Lemes, Jeffer E. Sasaki, Jorge Mota, Raphael M. Ritti-Dias, Luiz Carlos M. Vanderlei, Diego G. D. Christofaro

**Affiliations:** 1Department of Physical Education, School of Technology and Sciences, São Paulo State University (UNESP), Presidente Prudente 19060-900, Brazil; 2Department of Physiotherapy, Universidade Federal de Minas Gerais (UFMG), Belo Horizonte 31270-901, Brazil; 3Post-Graduate Program in Physical Education, Universidade Federal do Triângulo Mineiro (UFTM), Uberaba 38025-180, Brazil; 4Research Center on Physical Activity, Health and Leisure (CIAFEL), Faculty of Sport, Universidade do Porto, 4200-450 Porto, Portugal; 5Post-Graduate Program in Rehabilitation Sciences, Universidade Nove de Julho (Uninove), São Paulo 01525-000, Brazil

**Keywords:** physical activity, youth, motor behavior, public health

## Abstract

Introduction: Investigating the determinants of physical activity (PA) is an important strategy for the promotion of healthy lifestyles, mainly with PA of a moderate-to-vigorous intensity, which provides several health benefits in adulthood. In this sense, it is not clear whether early sports practice (ESP) during childhood and adolescence could be associated with the habitual practice of PA of higher intensities in adulthood. Objective: This study aimed to analyze the association of ESP in childhood and adolescence with different intensities of habitual PA in adulthood. Methods: A sample of 264 community-dwelling adults were randomly selected (42.2 ± 17.0 years, 57.5% of women). ESP during childhood and adolescence was evaluated using retrospective questions. Weekly minutes of PA were assessed using accelerometry and classified according to intensity as light, moderate, moderate-to-vigorous, vigorous and very vigorous. The association of ESP with a high level of PA (above median) in each intensity was analyzed using binary regression models. Results: The prevalence of ESP was 42.8% in childhood and 49.2% in adolescence. ESP in childhood was associated with a high level of very vigorous (OR: 2.48, *p* < 0.001) and vigorous PA (OR: 2.91, *p* < 0.001) in adulthood, but lost significance after adjustments by sex and age. ESP in adolescence was associated with a high level of very vigorous PA (OR: 1.99, *p* = 0.013) in the crude model and vigorous PA (OR: 2.21, *p* = 0.006), even after adjustments by age, sex and socioeconomic status. Conclusions: Engagement in sports practice during adolescence was associated with high levels of vigorous PA in adulthood and is an important period for healthy lifestyle promotion.

## 1. Introduction

Sports practice in youth has shown to improve several parameters of health, including cardiac autonomic control [1], bone mineral density [2,3], lower back pain [4], quality of life and sleep [5,6]. During childhood and adolescence, sports practice has been very popular and helps with the achievement of PA recommendations, as the World Health Organization recommends that children and adolescents perform at least 60 min of moderate-to-vigorous physical activity (PA) every day [7].

Previous studies have shown that children and adolescents who practice sports are more likely to have greater physical fitness in adulthood [8]. Adulthood is a period of life marked by significant reductions in PA levels [9], mainly of those with higher intensity, which has been inversely associated with age [10]. This is a public health problem as 27.5% of the global adult population have insufficient PA levels [11]. Adults with insufficient levels of PA are more likely to have several health impairments, such as poor cardiometabolic health [12], chronic diseases [13] and a higher risk of mortality [14].

It has been observed that early engagement in sports practice is associated with several health benefits in adult life, such as a lower prevalence of cardiovascular and metabolic outcomes [15], better mental health [16] and healthy lifestyle choices [17]. However, it is still unclear whether the engagement in sports practice during youth is associated with the practice of PA of a higher intensity during adulthood. The investigation of the determinants of PA in adult life is important to guide health promotion strategies focusing on healthy lifestyle consolidation throughout life stages. 

Therefore, this study aimed to analyze the association of early sports practice in childhood and in adolescence with different intensities of objective-measured PA in a randomly selected sample of community-dwelling adults.

## 2. Methods

### 2.1. Sample

This study sample consisted of community-dwelling adults in the city of Santo Anastacio, located in the southeast region of Brazil and with approximately 20,000 inhabitants. (Human Development Index (is a general and synthetic measure used to classify the degree of economic development and the quality of life of countries) = 0.79.) 

The random sample selection process considered the 23 urban census tracts, in which neighborhoods, streets and households were selected using the “random” function in the SPSS software, considering the proportion of inhabitants in each census tract for the proportion of sampling selection. The study protocol, sample size calculation and sampling selection process have been previously detailed in the literature [18]. A final sample of 264 participants (57.5% women; *n* = 152) was assessed. All participants who agreed to participate in the study signed a consent form. The present study was approved by the ethics and research committee of São Paulo State University (protocol CAAE: 72191717.9.0000.5402).

### 2.2. Early Sports Practice

Sports practice during childhood and adolescence was retrospectively assessed by the following questions: “When you were 7 to 10 years old, did you engage in any supervised sports activity for at least one uninterrupted year out of school (considering the holiday periods in the middle and end of the year)?” and “When you were 11 to 17 years old, did you engage in any supervised sports activity for at least one uninterrupted year of school (considering the holiday periods in the middle and end of the year)?” The answers for each question were “yes” or “no” [19,20]. 

### 2.3. Physical Activity Intensities

The different intensities of habitual PA in adulthood were measured using accelerometry. For this procedure, the participants were instructed to wear the accelerometers for 10 daily hours or more during at least five days. For definition of PA intensities in the accelerometer output, the cut-off points proposed by Sasaki et al. were used: light PA: 200 to 2689 counts per minute (cpm); moderate PA: 2690 to 6166; vigorous PA: 6167 to 9642; and very vigorous PA: 9642 or higher [21]. After determining the amount of time in each PA intensity, median value was used for definition of high level of PA (being above the median) in each intensity. Specifically for moderate-to-vigorous PA, the recommendations of the World Health Organization of 150 min of moderate-to-vigorous physical activity were considered [22].

### 2.4. Covariates

The variables of age, sex and socioeconomic status were considered as covariates in the present study. The socioeconomic status was assessed by Brazilian Criteria for Economic Classification [23], which considers the educational level of participants, the presence and quantity of specific rooms and consumer goods at home, in addition to urban infrastructure of the household and presence of housemaids. This instrument has specific scoring which classifies socioeconomic status into classes from highest to lowest (A, B1, B2, C1, C2, D–E).

### 2.5. Statistical Analysis

The sample characteristics were presented as mean and standard deviation, while categorical variables were presented in absolute and relative frequency. Chi-square test was used for the comparison of proportions, while independent sample *t*-test was used for mean comparison. The association between early sports practice in childhood and adolescence with different PA intensities was assessed using binary logistic regression. For this, three models were created (model 1: crude; model 2: adjusted by sex and age; model 3: adjusted by sex, age and socioeconomic status). The statistical significance was defined at *p* < 0.05 level and confidence interval at 95%. The statistical package used was SPSS version 24.0.

## 3. Results

The final sample of 264 community-dwelling adults presented a prevalence of early sports practice of 42.8% (n = 113) in childhood and 49.2% (n = 130) in adolescence. Table 1 shows the characteristics of the sample. Male participants showed a lower age than the females (39.8 ± 16.6 vs. 44.4 ± 17.2, *p* = 0.027), whereas no difference was observed in the body mass index (27.9 ± 5.3 for males vs. 28.5 ± 5.4 for females, *p* = 0.401). The prevalence of sports practice in childhood was 42.6% and in adolescence was 49.1% and was statistically different according to sex, where a higher proportion of males practiced sports in both childhood (67.3% vs. 24.3%, *p* < 0.001) and adolescence compared to the females (70.8% vs. 32.9%, *p* < 0.001). A total of 35.5% of participants (n = 94) reported having practiced sport in both childhood and adolescence, and this was higher in males than females (59.3% vs. 17.8%, *p* < 0.001). Regarding differences according to early sports practice, it was observed that participants who practiced sports in childhood and in adolescence were younger than those who did not practice sport in the same life stage (36.1 ± 15.1 vs. 47.2 ± 16.9 and 36.9 ± 15.1 vs. 47.8 ± 17.2, respectively), but no difference was observed in the body mass index.

Figure 1 shows the box and whisker plots of the weekly amount of PA intensities according to the early sports practice in childhood and in adolescence. Participants who reported to practice sports in childhood had higher weekly minutes of vigorous (*p* = 0.021) and very vigorous PA (*p* = 0.001) than those who did not practice sports in this life stage. A higher weekly time in vigorous-intensity PA (*p* = 0.005) and in moderate-to-vigorous PA (*p* = 0.034) was also observed in participants who reported to practice sports in adolescence when compared to those who did not. No significant difference was observed regarding the weekly minutes of light and moderate-to-vigorous PA according to early sports practice neither in childhood nor in adolescence.

Table 2 shows the association between early sports practice in childhood and the level of PA at different intensities in adulthood. The bivariate analysis showed that participants who reported to have practiced sports in childhood were more likely to have higher levels of vigorous (OR: 2.91, *p* < 0.001) and very vigorous PA (OR: 2.48, *p* < 0.001) than those who did not practice sports in this life state. However, when the analysis was adjusted by sociodemographic factors, no association remained significant (*p* > 0.05).

The association between early sports practice in adolescence with different intensities of PA is presented in Table 3. The bivariate logistic regression model showed that participants with sports practice in adolescence were more likely to have higher vigorous (OR: 3.78, *p* < 0.001) and very vigorous PA (OR: 1.99, *p* = 0.013) than those without sports practice in this life stage. In the analysis adjusted by sex, age and socioeconomic level, only the vigorous-intensity PA (OR: 2.21, *p* = 0.006) remained associated with early sports practice in adolescence.

## 4. Discussion

The present study aimed to verify the relationship between early sports practice and the different PA intensities in adulthood. As main findings, early sports practice in childhood was associated with vigorous PA in adulthood, but this association lost significance after adjustment by sex and age. On the other hand, early sports practice in adolescence was associated with vigorous PA regardless of age, sex and socioeconomic status. 

The prevalence of early sports practice in the present study was higher in males than females for both the childhood and adolescence periods. Studies have shown that sports practice tends to be different between boys and girls. Telford et al. [24] show in a study with children and adolescents aged 8–12 years that girls were less active than boys and participated less in sports offered by sports clubs. Considering the average age of our study, in which most participants were reporting sports practices from 30 years ago, and the fact that it is Brazil where the most practiced sport is soccer (and only recently has there been an advance in the practice of this sport by girls as well), these would be the factors to be considered. In addition, age is another factor to be mentioned, as the practice of physical activity tends to be lower with advancing age [25]. These factors may have had an influence on MVPA and on the loss of association between childhood sports practice and vigorous activity in adulthood.

This study observed that participants who reported to have early sports practice in adolescence were more likely to have a high level of vigorous-intensity PA in adulthood. Our findings are corroborated by Bélanger et al. [26]. In a study with 673 young Canadians, it was observed that the practice of sports in adolescence was associated with a greater practice of physical activity in adulthood. An interesting finding by Belanger et al. is that the greater the number of years of sports practice in adolescence, the greater the probability of remaining active in adulthood. The importance of continuing the practice of physical activity has been addressed in the literature. Telama et al. [27] in a cross-sectional study on the practice of physical activity in childhood and adolescence predict high levels of physical activity in adulthood. Batista et al. [28] in a systematic review observed that the practice of sports in childhood and adolescence was correlated with the practice of physical activity in adulthood.

Sports practice, for a large part, has times when greater cardiorespiratory and muscle action is required, often due to intermittent activities, for example, in sports such as soccer, basketball, handball, among others, which require a greater intensity of physical activity. This statement seems to be confirmed by the study of Sprengeler et al. [29] who observed that children involved in sports practice had higher physical activity intensities. Marques et al. [30] in a study with Portuguese children and adolescents observed that those involved in sports practice had greater chances of complying with the physical activity guidelines, as well as greater chances of engaging in vigorous physical activity. It is also noteworthy that in a secondary analysis of our study, participants who played sports in childhood and adolescence were more likely to play sports in adulthood as well. Such findings may help to explain the association between lifelong physical activity and greater intensities of physical activity. The maintenance of physical activity at moderate and high intensities tends to provide health benefits both in childhood and adolescence and should be encouraged.

This study observed no association between early sports practice and light PA in adulthood. This finding may be explained, at least in part, by the fact that light intensity PA is usually performed regardless of previous engagement in other forms of PA. For instance, adults engage in light PA when moving around, taking a shower, washing dishes, making the bed, among others. In our study, the average of light physical activity in adulthood is higher when compared to the other intensities. Such results were also observed for moderate physical activity. Factors such as age, sex and socioeconomic status are considered as adjustments in this study and may influence the results [31,32].

As limitations of the present study, the retrospective assessment of early sports practice by self-reporting may be inherent to memory bias, although previously validated simple questions were used. The lack of information about the weekly frequency and amount of early sports practice precluded to infer whether higher early sports practice could be associated with higher PA intensity in adulthood. Otherwise, as positive aspects, we highlight the random sample selection process in a community environment, considering the representativeness of the sample in the population of the city where the study was carried out, as well as the consideration of potential confounding factors in the statistical analysis. The objectively assessed PA intensities is another strength of the study, which precluded memory and classification bias by the participants, although it was not possible to infer whether more intense PA practiced in adulthood could be derived from the continuity of sports practice in adulthood or from other domains of adult life, such as occupation or commuting.

## 5. Conclusions

In summary, early sports practice was associated with a high level of greater intensity PA in adulthood, mainly when practiced in the adolescence period. It emphasizes the importance of sports practice promotion in younger populations and its continuity in further life stages.

## Figures and Tables

**Figure 1 ijerph-19-14753-f001:**
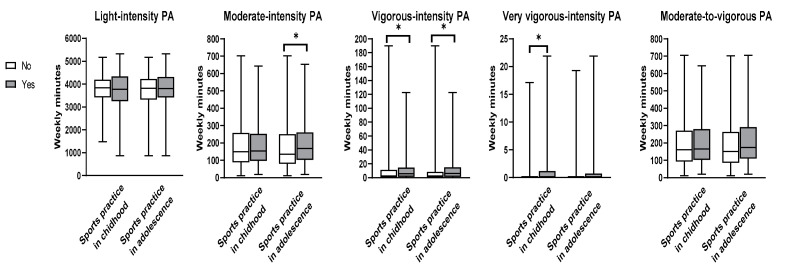
Physical activity intensities according to early sports practice in childhood and adolescence among community-dwelling adults (n = 264). * Statistical significance at *p* < 0.05 level.

**Table 1 ijerph-19-14753-t001:** Characteristics of the sample (n = 264).

	Mean	Standard Deviation
Age (years)	42.42	17.04
Weight (kg)	77.10	15.85
Height (cm)	165.60	9.77
BMI (kg/m^2^)	28.14	5.24
Light PA (min/day)	875.40	856.31
Moderate PA (min/day)	185.12	129.52
MVPA (min/day)	199.06	138.28
Vigorous PA (min/day)	12.48	23.23
Very vigorous PA (min/day)	1.11	3.13

min/day = minutes per day.

**Table 2 ijerph-19-14753-t002:** Association between sports practice in childhood with different intensities of physical activity in adult life.

	Sports Practice in Childhood		
	Odds Ratio	95%CI	*p*
		Light PA	
Model 1 (crude)	0.90	0.55–1.46	0.670
Model 2 (adjusted by sex)	0.76	0.44–1.31	0.328
Model 3 (adjusted by age)	1.01	0.60–1.70	0.958
Model 4 (adjusted by sex and age)	0.86	0.49–1.51	0.601
Model 5 (adjusted by sex, age and socioeconomic status)	0.85	0.49–1.51	0.597
		Moderate PA	
Model 1 (crude)	1.09	0.67–1.76	0.749
Model 2 (adjusted by sex)	0.84	0.48–1.45	0.529
Model 3 (adjusted by age)	0.89	0.53–1.51	0.682
Model 4 (adjusted by sex and age)	0.68	0.38–1.22	0.205
Model 5 (adjusted by sex, age and socioeconomic status)	0.69	0.38–1.23	0.211
		Vigorous PA	
Model 1 (crude)	2.91	1.76–4.82	<0.001
Model 2 (adjusted by sex)	2.02	1.16–3.53	0.012
Model 3 (adjusted by age)	2.25	1.32–3.84	0.003
Model 4 (adjusted by sex and age)	1.52	0.84–2.74	0.162
Model 5 (adjusted by sex, age and socioeconomic status)	1.49	0.83–2.69	0.185
		Very Vigorous PA	
Model 1 (crude)	2.48	1.44–4.26	<0.001
Model 2 (adjusted by sex)	1.90	1.05–3.45	0.034
Model 3 (adjusted by age)	1.87	1.06–3.31	0.032
Model 4 (adjusted by sex and age)	1.41	0.75–2.64	0.283
Model 5 (adjusted by sex, age and socioeconomic status)	1.41	0.75–2.64	0.287
		MVPA(≥150 min/week)	
Model 1 (crude)	1.58	0.96–2.60	0.072
Model 2 (adjusted by sex)	1.25	0.72–2.17	0.427
Model 3 (adjusted by age)	1.26	0.74–2.14	0.390
Model 4 (adjusted by sex and age)	0.98	0.54–1.76	0.954
Model 5 (adjusted by sex, age and socioeconomic status)	0.99	0.54–1.78	0.970

Model 1: unadjusted; Model 2: adjusted by sex; Model 3: adjusted by sex, age and socioeconomic level.

**Table 3 ijerph-19-14753-t003:** Association between sports practice in adolescence with different intensities of physical activity in adult life.

	Sports Practice in Adolescence		
	OR	95% CI	*p*
		Light PA	
Model 1 (crude)	0.99	0.61–1.59	0.952
Model 2 (adjusted by sex)	0.87	0.52–1.47	0.619
Model 3 (adjusted by age)	1.12	0.67–1.87	0.660
Model 4 (adjusted by sex and age)	1.00	0.57–1.72	0.989
Model 5 (adjusted by sex, age and socioeconomic status)	0.99	0.57–1.72	0.980
		Moderate PA	
Model 1 (crude)	1.41	0.87–2.29	0.158
Model 2 (adjusted by sex)	1.20	0.71–2.03	0.485
Model 3 (adjusted by age)	1.21	0.73–2.03	0.450
Model 4 (adjusted by sex and age)	1.04	0.59–1.79	0.898
Model 5 (adjusted by sex, age and socioeconomic status)	1.05	0.60–1.82	0.866
		Vigorous PA	
Model 1 (crude)	3.78	2.27–6.30	<0.001
Model 2 (adjusted by sex)	2.88	1.68–4.94	<0.001
Model 3 (adjusted by age)	3.03	1.78–5.15	<0.001
Model 4 (adjusted by sex and age)	2.28	1.30–4.01	0.004
Model 5 (adjusted by sex, age and socioeconomic status)	2.21	1.25–3.92	0.006
		Very Vigorous PA	
Model 1 (crude)	1.99	1.15–3.40	0.013
Model 2 (adjusted by sex)	1.51	0.84–2.70	0.169
Model 3 (adjusted by age)	1.46	0.82–2.59	0.194
Model 4 (adjusted by sex and age)	1.10	0.59–2.04	0.764
Model 5 (adjusted by sex, age and socioeconomic status)	1.09	0.59–2.03	0.792
		MVPA (≥150 min/week)	
Model 1 (crude)	1.80	1.10–2.94	0.020
Model 2 (adjusted by sex)	1.50	0.88–2.55	0.132
Model 3 (adjusted by age)	1.46	0.87–2.47	0.152
Model 4 (adjusted by sex and age)	1.22	0.70–2.13	0.482
Model 5 (adjusted by sex, age and socioeconomic status)	1.24	0.70–2.18	0.454

OR: odds ratio; CI: confidence interval; PA: physical activity; MVPA: moderate-to-vigorous physical activity.

## Data Availability

The data presented in this study are available upon reasonable request from the corresponding author. The data are not publicly available due to ethical commitments for sensitive patient information.

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
