# Peer review of "Can Sports Practice in Childhood and Adolescence Be Associated with Higher Intensities of Physical Activity in Adult Life? A Retrospective Study in Community-Dwelling Adults"

_ijerph, 2022, doi:10.3390/ijerph192214753_

Round 1
Reviewer 1 Report
Well, this is what I would call a " well very nice " research which basically is common sensical---that individuals who engage in sports- be it recreational or formal team events tend to remain active, and tend to enjoy physical activities later in their life. The statistics are all good and appropriate. How earth shattering the results are----that is another thing- I see this paper as a " filler"- if the Editor has space and needs an article to complete the journal this is great.
How significant is this in terms of content- I gave this a LOW- it is basically good common sense, borne out by reseach.
Author Response
Dear Editor and Reviewers,
We would like to thank for the opportunity to resubmit the attached manuscript entitled “Can sports practice in childhood and adolescence be associated with higher intensities of physical activity in adult life? A retrospective study in community dwelling adults” for possible publication in the International Journal of Environmental Research and Public Health.
The authors also wish to express their gratitude for the efforts of the editor and reviewers in directing the manuscript towards a more acceptable form for publication. The manuscript has been carefully checked, and appropriate changes made following the reviewers’ suggestions.
All the revisions in the manuscript were marked using the “Track Changes” function of MS Word and we have attached a separate document containing point-by-point responses to the reviewer comments.
The authors hope that the added revisions adequately address the comments and we are available for any other queries.
Sincerely,
The authors
Reviewer 1
Well, this is what I would call a " well very nice " research which basically is common sensical---that individuals who engage in sports- be it recreational or formal team events tend to remain active, and tend to enjoy physical activities later in their life. The statistics are all good and appropriate. How earth shattering the results are----that is another thing- I see this paper as a " filler"- if the Editor has space and needs an article to complete the journal this is great.
How significant is this in terms of content- I gave this a LOW- it is basically good common sense, borne out by reseach.
Response: Dear reviewer, we are grateful for the consideration of our manuscript. Although several studies have shown that children and adolescents who practice sports at this stage of life are more likely to practice physical activity in adulthood, our study go forward investigating whether early sports is associated with more intense physical activity, which is not clear in current literature. We considered that this was an important finding, since moderate to vigorous physical activity provide several health benefits in adulthood and early sports can be an important factor for promotion of higher amount of moderate-to-vigorous physical activity in further life stages.

Reviewer 2 Report
Line 42: “The adulthood”, replace with “Adulthood”
Line 46: "27.5% of the global adult population have insufficient PA levels"
Line 51: "it is still unclear"
Line 62: Need to explain the Human Development Index.
Line 63: "The random sample"
Lines 74-79: Is the question assuming that the participant played a sport for the entire school year, if they played fall and winter but not spring, should their answer be no?
Lines 81-82: Need to reword "accelerometer, e.g., accelerometry or accelerometers were used to measure PA"
Line 87: "median value was used"
Line 155: Why not split the sex and age model? This would give more understanding within each category. Combining the two can still be evaluated as well.
Line 168: , aged 8-12 years, that girls were
Line 186: Batista et al. needs to be cited again.
Line 190: Is football referring to soccer?
Concluding Thought
This was a solid paper that gives evidence to the importance of sport before adulthood. Which brings value to the field and society.
Author Response
Dear Editor and Reviewers,
We would like to thank for the opportunity to resubmit the attached manuscript entitled “Can sports practice in childhood and adolescence be associated with higher intensities of physical activity in adult life? A retrospective study in community dwelling adults” for possible publication in the International Journal of Environmental Research and Public Health.
The authors also wish to express their gratitude for the efforts of the editor and reviewers in directing the manuscript towards a more acceptable form for publication. The manuscript has been carefully checked, and appropriate changes made following the reviewers’ suggestions.
All the revisions in the manuscript were marked using the “Track Changes” function of MS Word and we have attached a separate document containing point-by-point responses to the reviewer comments.
The authors hope that the added revisions adequately address the comments and we are available for any other queries.
Sincerely,
The authors
Reviewer 2
Line 42: “The adulthood”, replace with “Adulthood”
Response: The change has been made.
Line 46: "27.5% of the global adult population have insufficient PA levels"
Response: Done.
Line 51: "it is still unclear"
Response: Done.
Line 62: Need to explain the Human Development Index.
Response: The Human development index is a general and synthetic measure used to classify the degree of economic development and the quality of life of countries. We inserted this information into the methods.
Line 63: "The random sample"
Response: Done
Lines 74-79: Is the question assuming that the participant played a sport for the entire school year, if they played fall and winter but not spring, should their answer be no?
Response: In the region of Brazil where the study was carried out, we do not have severe winters, which could prevent or reduce the practice of physical activity. However, if the participant practiced the sport in most months of the year, the answer should be yes.
Lines 81-82: Need to reword "accelerometer, e.g., accelerometry or accelerometers were used to measure PA"
Response: We thank for the comment. The word “accelerometer” was replaced by “accelerometry”.
Line 87: "median value was used"
Response: Done.
Line 155: Why not split the sex and age model? This would give more understanding within each category. Combining the two can still be evaluated as well.
Response: Dear reviewer, we appreciate the comment. We have presented the statistical analysis into five models, being Model 1 (crude), Model 2 (adjusted by sex), Model 3 (adjusted by age), Model 4 (adjusted by sex and age), and Model 5 (adjusted by sex, age, and socioeconomic status).
Line 168: , aged 8-12 years, that girls were
Response: Done.
Line 186: Batista et al. needs to be cited again.
Response: Done.
Line 190: Is football referring to soccer?
Response: Yes, we corrected the nomenclature in the text. Thanks for the comment.
Concluding Thought
This was a solid paper that gives evidence to the importance of sport before adulthood. Which brings value to the field and society.
Response: Thank you very much for the comment.
